# Review on Polysaccharides Used in Coatings for Food Packaging Papers

**Petronela Nechita [1],* and Mirela Roman (Iana-Roman) [2]**

[1]   Department of Environmental, Applied Engineering and Agriculture, Faculty of Engineering and Agronomy, Dunărea de Jos University of Galați, 817112 Brăila, Romania

[2]   Doctoral School of Fundamental and Engineering Sciences, Dunărea de Jos University of Galați, 817112 Brăila, Romania; mirela.roman@ugal.ro

*   Correspondence: petronela.nechita@ugal.ro; Tel.: +40-239-744704928

**Abstract:** Paper and board show many advantages as packaging materials, but the current technologies employed to obtain adequate barrier properties for food packaging use synthetic polymers coating and lamination with plastic or aluminium foils—treatments which have a negative impact on packaging sustainability, poor recyclability and lack of biodegradability. Recently, biopolymers have attracted increased attention as paper coatings, which can provide new combinations in composite formulas to meet the requirements of food packaging. The number of studies on biopolymers for developing barrier properties of packaging materials is increasing, but only a few of them are addressed to food packaging paper. Polysaccharides are viewed as the main candidates to substitute oil-based polymers in food paper coating, due to their film forming ability, good affinity for paper substrate, appropriate barrier to gases and aroma, and positive effect on mechanical strength. Additionally, these biopolymers are biodegradable, non-toxic and act as a matrix for incorporation additives with specific functionalities for coated paper (i.e., active-antimicrobial properties). This paper presents an overview on the availability and application of polysaccharides from vegetal and marine biomass in coatings for foods packaging paper. The extraction methods, chemical modification and combination routes of these biopolymers in coatings for paper packaging are discussed.

**Keywords:** polysaccharides; food packaging paper; coating; hemicelluloses; chitosan; cellulose derivatives

## 1. Introduction

Food packaging coatings are very important, as they provide a physical protection barrier for food products during storage and transportation, maintain food safety in a way that satisfies industry requirements and consumer desires, and minimize environmental impact by reducing food waste. Paper based materials have been used for packaging of fluids and greasy foods since the 1880s. During the 1970s–1980s, when plastics were introduced into food packaging, paper-based materials lost their importance, being replaced in many uses. Recently, packaging trends are dominated by sustainability action, largely fuelled by rising anti-plastic sentiment, and studies are focused on green packaging based on renewable resources that are recyclable, biodegradable and/or compostable [1]. The attention on environmentally friendly alternatives prompted the food packaging industry to search for an alternative reliable and sustainable coating product, both for support materials and for polymeric ones. From this perspective, paper appears as the ideal packaging material, having the advantages of high recyclability, biodegradability and compostability from renewable raw material, in comparison with petroleum based packaging [2].

As food packaging material, due to its porous structure and the hydrophilic character of cellulose fibers, paper/paperboard has inherently poor barrier properties (i.e., low water and grease resistance, high permeability to gases and water vapors) and is sensitive to microbial attack.

In order to fulfil protection requirements, packaging paper should provide different barrier properties a (i.e., against oxygen, carbon dioxide, moisture, water, micro-organisms, grease, or aroma) in relation to the packaging end-uses. At the end of 19th century, several technologies were developed for food packaging paper with good barriers against the grease and gases, such as: greaseproof and glassine paper based on intensive mechanical treatments of cellulose fibers, and parchment paper made by running paper through a sulphuric acid bath. These technologies are energy-intensive and have become unsustainable with the steady increase of energy costs during the last decades. During the 20th century, different food packaging paper grades were developed based on coatings with waxes/oil-polymers and on lamination with aluminium and plastic foils. These treatments have a negative impact on packaging sustainability and are based on petroleum-derived polymers, which increase the carbon footprint of the packed product and affect the recyclability and biodegradability of the used packages [3,4].

The development of coatings from natural polymers for food packaging applications has been a high interest topic for several years, due to increasing environmental concerns and the prices of petrochemicals. For more natural products, improving the quality of bio-based films or biopolymers is an important step to satisfy consumers demand of more environmentally friendly packaging. This approach will continue to play an important role in the food industry.

Due to the environmental concerns and the realization that global petroleum resources are finite, biopolymers are attracting increased attention as they not only can replace existing polymers in different applications but also provide new combinations of properties.

At present, the number of studies on biopolymers for developing barrier properties of packaging materials is increasing continuously, but only a few of them are addressing applications for food packaging paper [5].

The biopolymers studied to develop/improve barrier properties of packaging materials, including paper-based ones, are grouped as: (i) polysaccharides (chitosan, starch, lignocellulose derived compounds, alginates); (ii) proteins (whey, wheat gluten, and zein); polyesters (polylactic acid (PLA), polycaprolactone (PCL), and polyhydroxyalkanoates (PHAs) [6].

PLA is the most studied biopolymer for food contact packaging, including paper based ones. However, it is still a challenge to produce packaging paper coated with PLA that is competitive with oil-based polymers [7] as PLA is thermally unstable and degrades during processing, due to hydrolysis and chain scission. Additionally, the use of thermal stabilizers and plasticizers to improve PLA processing reduces its biodegradability [8].

The polysaccharides are viewed as the main candidates to substitute oil-based polymers in food paper coating, as they are: biodegradable and non-toxic; have film forming ability and good affinity for paper substrate; can provide a very good barrier to gases, aromas, and lipids; and have positive effects on mechanical strength [6].

In this paper, we present a systematic overview on the availability of polysaccharides from vegetal and marine biomass and their performances in functional coatings for foods packaging paper. In addition, the extraction methods, chemical or physical modification routes, combination with other compounds and the environmental impact of these biopolymers when used in paper packaging coating applications are discussed.

## 2. Polysaccharides in Food Packaging Paper

Polysaccharides are carbohydrates polymers consisting of repeating units of monosaccharides (hundreds and thousands) linked by glycosidic bonds and formed by the condensation of monosaccharide residues through hemiacetal or hemiketal linkages. The polysaccharides can originate from higher photosynthetic plants, marine biomass, bacteria or fungi [9–12]. At the cellular level,

polysaccharides represent the reserve compounds in cytoplasm, or structural components of the membrane and cell wall of organisms [13,14]. They are biocompatible, biodegradable and non-toxic towards living organisms. These characteristics give them the potential for broad applications such as: medicine, drugs delivery and food packaging. In food packaging, polysaccharides have the potential to be used as coating formulations for paper, edible coatings and films or to obtain bioactive and sensor materials in active and intelligent packaging [15,16]. The most tested and applied polysaccharides in the paper industry, that include coated paper for foods packaging, are presented in Figure 1.

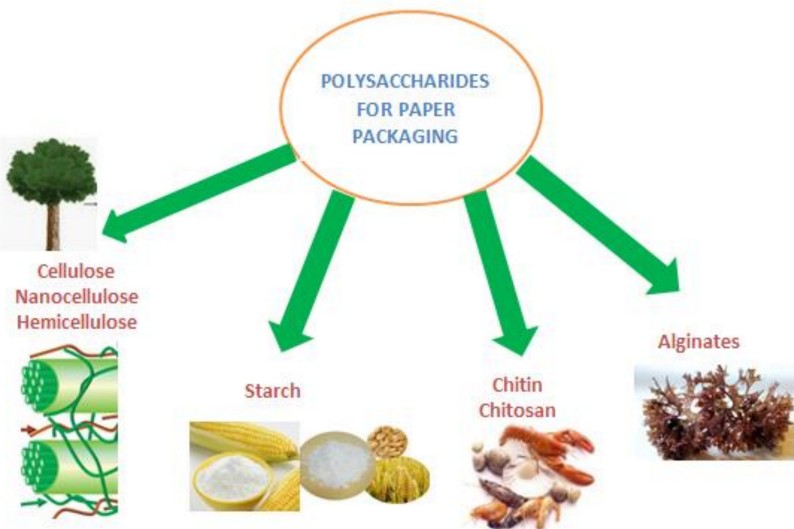

**Figure 1.** Polysaccharides used in paper industry and their origin.

When used in coatings or films for food packaging, polysaccharides present some drawbacks, which are linked with their hydrophilic nature and crystalline structure, i.e., low water resistance, poor barrier to water vapors, and properties dependence on the environment humidity. However, to achieve the molecular structures of interest it is very important that polysaccharides be chemically modified and functionalized. In recent years, this has became a research topic and area that has been extensively studied. In addition, many fibers, other biopolymers and environmentally friendly nanomaterials are combined with polysaccharides to prepare novel formulations with desired properties for food packaging, including coatings for paper/board packaging. Sometimes it is necessary to have a combination of more than two packaging materials to provide the best packaging solution for certain food products [17,18].

*2.1. Polysaccharides from Wood and Lignocellulosic Plants*

2.1.1. Cellulose and Cellulose Derivatives

Cellulose is a renewable and biodegradable resource, being the most abundant biopolymer in nature, that acts as consolidation component of plants and bacteria [19]. It is a linear polysaccharide consisting of repeated units of cellobiose, which is a combination of two anhydroglucose rings linked via a β-1,4 glycosidic bond (Figure 2) [20].

**Figure 2.** Chemical structure of cellulose.

In the supramolecular structure of cellulose two domains are displayed: a crystalline region with a high ordered structure and an amorphous region with a less ordered structure. Usually, the crystallinity degree of cellulose is about 40–60%, being influenced by the source and pretreatment applied to the cellulose sample. The hydroxyl groups located in the amorphous regions are accessible and highly reactive. Those present in the compact crystalline regions with highly cohesive energy exhibit a much lower accessibility. Interactions between solid cellulose and water, enzymes, or reactive substances occur first at the amorphous domains or at the surface of cellulose crystals. Due to the hydrogen bonds network, cellulose is a relatively stable polymer, with fibrous structure which does not readily dissolve in typical aqueous solvents and has no melting point [14,21,22].

Cellulose based materials and chemicals are used in various applications in the paper or textile industries. In the paper industry, cellulosic fibers are the main raw material for paper and board production, but cellulose and their derivatives can also used as coating materials for paper applications, including packaging.

Due to its hydrophilic character, water insolubility, poor film-forming ability and high crystallinity, the cellulose cannot be used in its native form. The cellulose derivatives such as carboxymethyl cellulose (CMC), methyl cellulose (MC), ethyl cellulose (EC), hydroxypropyl and hydroxyethyl cellulose (HPC and HEC) and hydroxypropyl methyl cellulose (HPMC) have been commercially produced to overcome these drawbacks. These products can be used for both wet-end and surface finishing of papers to improve barrier properties [23,24].

In food packaging, cellulose derivatives have been used since the early 1900s. Cellulose and their derivatives have the capability to mechanically reinforce and enhance the barrier properties of polymer materials. In addition, cellulose derivatives are more resistant to microbial attacks and enzymatic cleavage than native cellulose [25].

- Cellulose Ethers

Cellulose ethers are compounds with a high molecular weight produced by substituting the hydrogen atoms of hydroxyl groups in the anhydroglucose units with alkyl groups. The important properties of these cellulose derivatives include their solubility, viscosity in solution, surface activity, thermoplastic film characteristics and stability against biodegradation, heat, and that their hydrolysis and oxidation are influenced by their molecular weights, chemical structure and the distribution of substituent groups. The mostly used cellulose ethers are: MC, EC, HEC, HPC, HPMC and CMC [26].

CMC is frequently used as a co-binder and/or thickener in pigment coating color, but these derivatives have some limitations when used as a unique binder, so that the most common applications are as rheology modifiers or water holding agents [27,28].

MC, HPMC and HPC are biodegradable thermoplastic polymers that are soluble in cold water, and which form a hard gel after heating at 50–80 °C. HPMC is an edible material with good film-forming properties, which is odorless, flavorless, transparent, stable, oil-resistant and nontoxic. HPMC has a good miscibility with a wide range of organic and inorganic materials, and can be used as film-forming material and to control the barrier and mechanical properties in paper coatings. The water resistant, thermal and mechanical properties and film flexibility of this polymer can be improved by its combination with other polymers—plasticizers or inorganic nanoparticles (i.e., potato or corn starch, glycerol, graphene oxide) [29–31].

For example, the water vapor permeability (WVP) and water absorption capacity of coated paper with HPMC without plasticizers is reduced by 25% in comparison with uncoated paper. In contrast, an increasing of WVP and better coating flexibility is obtained when plasticizers are used in HPMC coatings. In addition, HPMC is used in coating colors to control the barrier and mechanical properties of coated papers. Furthermore, the barrier properties and smoothness of coated papers with HPMC can be improved by the addition of beeswax [32,33].

- Cellulose Esters

These polymers are water insoluble with good film forming properties and are often used in combination with the cellulose ethers to obtain the microporous membranes. Two groups of cellulose esters are used in different applications: organic (e.g., cellulose acetate) and inorganic groups (cellulose nitrate and cellulose sulphate) [26]. Organic cellulose esters (e.g., cellulose acetate, cellulose acetate phthalate, cellulose acetate butyrate, hydroxipropylmethyl cellulose phthalate) have been used in commercial or in pharmaceutical applications. Inorganic cellulose esters (e.g., cellulose nitrate) are transparent compounds with good film forming abilities, but which are rarely applied alone due to their very low solubility and high flammability. Cellulose acetate has been receiving attention for use in food contact applications due to its nontoxicity, edibility and biocompatibility. The most common applications of cellulose acetate are as films or fibers. In paper packaging applications cellulose esters are used in the paper lamination process. For example, cellulose acetate foils can be attached to the surface of paper or paperboard stock by adhesives or hot laminating either alone or combined with aluminium foils to obtain foods packaging. Using the electrospinning process cellulose acetate nanofibers have been obtained, with potential applications in packaging materials for fresh fruits and vegetables [34–36].

Due to the highly crystalline structure of cellulose the derivatization process is difficult and expensive, however based on good film forming and binding properties, cellulose derivatives are promising to use in the production of attractive biodegradable and functional materials for application in food packaging (i.e., membranes, edible films or paper coatings) [37,38].

- Cellulose Micro(nano)fibrillated Structures

The obtainment of micro(nano)-scale cellulose fibers and their application to produce added-value products (biocomposites for medicine, electronics, foods and foods packaging) has gained increasing attention due to their unique properties: high strength and stiffness, low weight, biodegradability and renewability.

In plant tissue the cellulose molecules are brought together into structural units known as elementary fibrils or microfibrils. These structural elements are packed as microfibrillated cellulose (MFC). The diameter of elementary fibrils is about 5 nm and microfibrillated cellulose, also called nanofibrillated cellulose (NFC), has diameters from 20 to 50 nm. This explains the existence of two classes of nanocellulose: (i) cellulose nanocrystals and (ii) cellulose microfibrils.

Nanocellulose has been around since the early 1980s and can be extracted from natural sources (i.e., cellulose from wood, lignocellulosic waste, agricultural and foods waste etc.) using chemical, enzymatic and mechanical processes, which include grinding and refining treatments [39].

Generally, to obtain NFC and cellulose nanocrystals, as main raw material, cellulose fibers are pre-treated by milling, pulping and bleaching to remove the non-cellulosic components, such as lignin and hemicellulose [40]. After that, the acid hydrolysis process using strong acids such as sulfuric acid is applied to destroy the amorphous domains and the individual nanocrystals in the form of a needle are released [41]. Mechanical treatments in high-pressure homogenizers produce the delaminating of cellulosic fibers. The diameter of fully delaminated nanocellulose fibers is in the range of 10–100 nm, and the length can be between few hundred nanometers to some microns [42]. In appearance, these nanofibrillar structures form a highly viscous and shear-thinning transparent gel (Figure 3). This is due to the strong increase in specific surface area of the fibers in the fibrillation process [43].

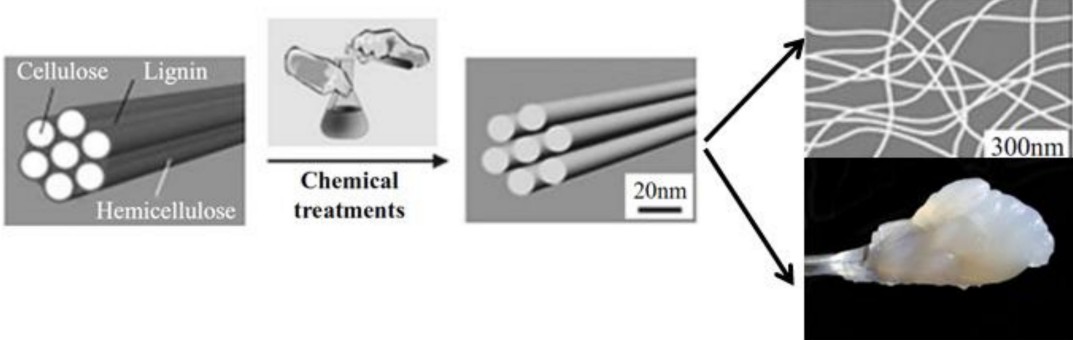

**Figure 3.** Procedure for individualizing cellulose nanofibers and nanocellulose suspension.

The films prepared using NFC suspensions exhibit properties depending on their fibrillation degree. Thus, a high-density packing of cellulose nanofibers with low porosity is obtained that results in optically transparent films. Another source of nanocellulose is bacterial cellulose. These cellulose nanofibers are secreted by specific bacteria (strains of *Gluconacetobacter*, *Komagataeibacter*, tea fungus) and have a width of about 3.5 nm and very good physical and mechanical properties (high elastic modulus, high specific surface area) and high purity [44,45]. Nowadays, bacterial cellulose is a topic of high investigation, with applications in numerous areas: biomedicine, reinforcement in nanocomposites, electronic papers, textiles, packaging or foods [41,46–48]. However, only a few studies have focused on the utilization of bacterial cellulose in the paper industry or in coatings for paper packaging. Some research studies are related on the use of bacterial cellulose in paper restoration [49] or to improve the barrier properties of paper products [50]. Based on these reasons, and as their industrial application does not seem impending, we have decided not to present any more details on this topic.

NFC exhibits interesting properties that make it attractive for many applications, including paper and board manufacturing. In regards to the paper and board industry, NFC could be used as a strengthening additive in paper with high filler content. In paper food packaging, NFC can be used as barrier material (against oxygen, water vapor, grease/oil) in surface paper treatments by sizing and coating. Other applications are in the field of nanocomposites, as thickeners in food industry, cosmetic/pharmaceutical and medicine applications, as well as in the electronics sector [39,42]. According to the literature review, the microfibrillated cellulose structures are composed of nanofibrils, fibrillar fines, fiber fragments and fibers (Table 1) [51].

**Table 1.** Components of microfibrillated cellulose structures.

| Diameter (μm) | Biological Structure | Technological Terms |
|---|---|---|
| 10–50 | Tracheid | Cellulose fibers |
| <1 | Macrofibrils | Fibrila fines, fibrils |
| <0.1 | Microfibrils Elementary fibril | Nanofibrils, nanofibers |

Due to their hydrophilic nature, cellulose micro(nano)fibrillated structures can be chemical modified by reactions of sulfonation, carboxylation, grafting or reactions that create hydrophobic surfaces (acetylation, silanization treatments) or adsorption (surfactants, polyelectrolytes). Via these treatments ionic groups are formed, by the adsorbing of hydrophobic compounds to surface hydroxyl groups of the cellulosic nanoparticles [52]. After chemical modification by silylation, the wettability of the NFC film surface was drastically reduced and extended through the whole film and as a result its dimensional stability when submerged in water was significantly improved [53].

When associated with the other cellulosic materials or when used in nanocomposite applications, cellulose micro(nano) fibrillated structures show good barrier characteristics. These functions,

developed as a response to the new and evolving societal requirements, extend their utilization in sectors such as the food industry and packaging [35].

Based on the compact structure formed by the cellulosic microfibrils as well as their ability to form intra- and inter-fibrillar hydrogen bonds, these cellulose derivatives can be promising for applications as transparent and biodegradable packaging films with high barrier properties (i.e., oxygen permeability lower than plasticized starch and whey protein). However, many studies have shown a high interest in applying MFC and NFC to improve the mechanical and barrier properties of food packaging and in the printing processes.

In their studies, Lavoine et al. [54] propose three strategies to asses and develop the barrier properties of MFC structures: (a) introduction of MFC in nanocomposites; (b) use of MFC as coating additive; and (c) use of MFC as 100% films.

Based on their low permeability against air, grease, and oxygen, which can be advantageous as a barrier layer in food packaging, MFCs have been studied as an alternative for barrier packaging films [54–57]. At relative humidity lower than 70%, cellulose nanomaterials have good oxygen barrier and poor moisture [58]. By optimizing the morphology and surface chemistry of cellulose nanomaterials, or sandwiching them with high moisture-resistant polymers, the barrier performance can be additionally improved [57].

The water vapor and moisture resistance of cellulose nanomaterial-based packaging films can be improved also, using different technologies such as: layer-by-layer assembly, electrospinning, composite extrusion, casting evaporation, coating, and all-cellulose composites [59].

In recent studies, Yook et al. [60] has been investigating the influence of the coating weight and the properties of NFC (morphology, surface chemical modification of nanofribrils) on the barrier against air, water, water vapor, oxygen and grease of coated papers with NFC gels. Based on the obtained results, a low coat weight cannot provide barrier properties due to insufficient coverage and thickness. A coat weight of 10 g/m$^2$ seems to be optimal for improving the barrier performance against air, liquid water, water vapor, and grease (Figure 4). Additionally, a smaller NFC diameter is advantageous for reducing the permeation of substances such as water vapor and oxygen, and the hydrophobization of the NFC surface is beneficial to improve the water barrier properties. The grease resistance is mainly influenced by the average fibril diameter which can be narrow at a certain value, while the functional group of the hydrophobic chemical affects the water vapor transmission rate and barrier to air and oxygen [60].

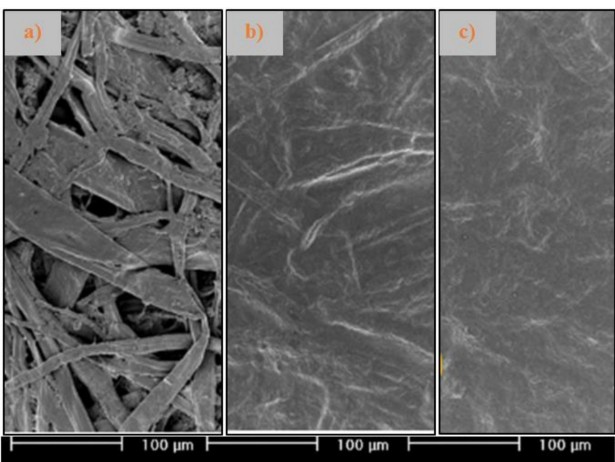

**Figure 4.** SEM images of papers coated with nanofibrillated cellulose (NFC) at different coat weights: (**a**) base paper; (**b**) NFC at 6 g/m$^2$; (**c**) NFC at 10 g/m$^2$.

The coat weight is correlated with the paper coating method which is very important to obtain the properties of the final products. For example, using the bar coating method a high coat weight (7 g/m$^2$) of MFC is obtained, while by size press and with multiple layers rarely reached at 4 g/m$^2$. As a result,

the improved air permeability (70% reduction) of MFC coated papers is obtained using bar coating application. When a similar MFC coat weight was applied on the paperboard surface, insignificant improvements in barrier properties were observed due to the high weight of the paperboard and relatively insufficient coat weight of the MFC [61,62].

Other recent studies [63] show that the barrier properties of NFC films are improved by hydrophobization of fibrils surface using sol–gel methods. By sol–gel methods, coatings are relatively dense and homogeneous coating layers are obtained with strong adhesion leading to decrease of surface hydrophilicity and water vapor transmission rate. Thus, the value of contact angles varied between 54° and 102° and the water vapor transmission rates between 230 and 410 $g/m^2$/day. Furthermore, at 80% RH the sol–gel coatings significantly improve the oxygen barrier properties. Thus, at 50% RH the water transmission rate varied from 0.4 to 0.5 $cc/m^2$/day and at 80% RH between 51 and 86 $cc/m^2$/day.

To improve the water resistance and barrier properties, in parallel with the introduction of the other specific functionalities, several other NFC-based paper coatings have been developed. However, besides their biodegradability, biocompatibility and availability, NFC has increased interest for use in the obtaining of active packaging, due to the polymer matrix ability to embed the active and antimicrobial additives. This provides an alternative with high quality, and many properties such as antimicrobial and harmful detection properties and a variety of additional barrier properties that enhance the final food package functionality. Improving the antibacterial properties of cellulose based materials by facile methods attracts the attention of many researchers. However, in the case of cellulose based packaging the antimicrobial functionalities can be obtained at different levels of manufacturing: (i) antimicrobial additives can be added to the cellulose fibers during and after the web forming (by addition in cellulose fibers suspension or by surface treatments of paper using surface sizing and coating) and (ii) antimicrobial components can be added to the finished paper product during converting processes by spraying or printing on the paper surface [64].

There are a large variety of active-antimicrobial materials which can be combined with NFC, but only a limited number of these have been investigated for antimicrobial paper. These include inorganic materials (e.g., metal oxides: titanium dioxide ($TiO_2$), zinc oxide (ZnO) and magnesium oxide (MgO), silver, gold or copper nanoparticles) and organic compounds (polysaccharides based on hemicelluloses, chitosan and chitosan derivatives), as well as biomolecules (nisin) [65–67].

In their research, Amini et al [67] obtained packaging papers with antibacterial properties using cellulose nanofibers impregnated with silver nanoparticles (NFC/Ag). These coatings were applied as thin layer on two base papers: kraft and greaseproof paper. The coatings applied on kraft paper showed resistance to water vapor transmission and water absorption, in spite of the hydrophilic nature of NFC. When was applied on greaseproof paper the coating not improved water absorption resistance. This difference can be attributed to water sorption mechanism on those two substrates. The obtained results show that using NFC/Ag coatings on kraft and greaseproof paper substrates can improve their potential to be used as environmentally friendly antimicrobial packaging for solid and oily foods.

### 2.1.2. Hemicelluloses

- General Features

The hemicelluloses (HCs) are the second most abundant plant polysaccharides after cellulose in the cell walls of lignocellulosic biomass, representing 20–35% of this, as a function of biomass source (Figure 5) [68]. Unlike cellulose, in which the monomer units are chemically homogenous, hemicelluloses are a series of complex, branched and heterogeneous polymers. Hemicelluloses are composed from sugar units, arranged in different proportion with various substituents and are called pentosans and hexosans. The main sugars are ᴅ-xylose, ʟ-arabinose, ᴅ-glucose, ᴅ-galactose, ᴅ-mannose, ᴅ-glucuronic acid, 4-O-methyl-ᴅ-glucuronic acid, ᴅ-galacturonic acid, and to a lesser extent, ʟ-rhamnose, ʟ-fucose, and various O-methylated neutral sugars. Hemicelluloses of cereal straws have a backbone of (l→4)-linked β-ᴅ-xylpyranosyl units Their relative content and structural composition also varies within a species depending on the location in the plant or cellular tissue origin.

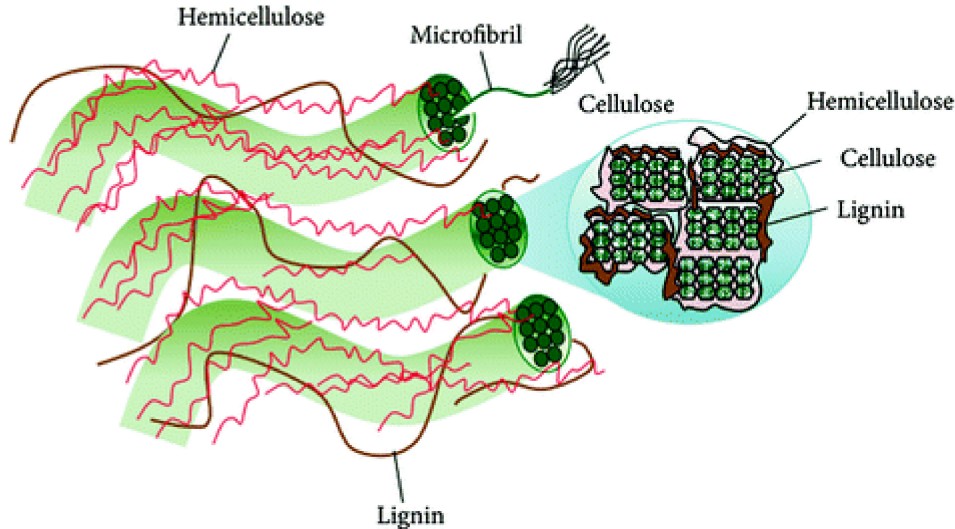

**Figure 5.** Arrangement of hemicellulose in the plant cell walls.

Hemicelluloses are abundant in agricultural by-products and woody materials. Cereal straws comprise the major portion of plant materials, and HCs amount is according to the particular plant species, such as: maize steins (28.0%), barley straw (34.9%), wheat straw (38.8%), rice straw (35.8%), and rye straw (36.9%) [69–71].

Xylans are the most abundant type of hemicelluloses in the cell wall of hardwoods, having xylopyranosyl backbone units substituted at C-2 with $\alpha$-(1→2)-linked 4-O-methyl-glucuronic acid residues (Figure 6); whereas, occurring in about 70% of the xylopyranose backbone is a hydroxyl group that is acetylated at C-2 or C-3 [72–74].

**Figure 6.** The chemical structure of xylan hemicellulose.

Glucomannan is major hemicellulose component of softwoods. Their chemical structure has a main chain of $\beta$-(1→4)-linked D-mannose and D-glucose with $\alpha$-(1→6)-linked D-galactose moieties in various amounts (Figure 7).

**Figure 7.** The chemical structure of glucomann hemicellulose.

- Extraction of HCs

The hemicelluloses can be separated within kraft-based dissolving pulp production process, or the concept of integrated lignocellulosic biorefinery and by treatment of biomass (wood, plants or lignocelluloses waste) using various extraction and isolation methods.

The extractability of HCs depends on the source and origin of biomass. Differences in hemicellulose extractability are determined by the plant-specific hemicellulose arrangement and lignin within the cell wall. The total lignin content of biomass is the main factor with high influence for HCs extractability [75].

The hemicelluloses are commonly removed during the initial stage of biomass processing aiming to reduce the structural complexity for further enzymatic cellulose hydrolysis. Various pre-treatment processes are available to fractionate, solubilize, hydrolyze and separate cellulose, hemicellulose and lignin. These include concentrated and dilute acid, alkaline and hydrogen peroxide extraction, steam explosion (autohydrolysis), ammonia fiber explosion, wet-oxidation extraction with hot water, $CO_2$ explosion, organic solvent treatments, enzymatic hydrolysis, ionic liquids and supercritical fluids, microwave and others [76–78].

Alkaline extraction is well studied approach, and it is known as a strong and efficient method for hemicellulose extraction. Nevertheless stepwise alkaline extraction methods are rarely studied. In their research, Li et al. [79] found that the relatively high alkali concentration dissolve straight chain type of hemicellulose, while relatively low alkali concentration dissolved branched chain type of hemicelluloses. Furthermore, the content of arabinose, glucose, and galactose presented in small amounts, and mannose were identified as minimal quantities. The optimal alkali concentration of extraction process was found to be at 1.5% KOH with the higher hemicellulose yield, xylose content and thermal stability of the hemicelluloses, at which the alkaline extraction process did not result in obvious changes in the macromolecular structure of hemicelluloses [79].

Moreover, it has been reported that the addition of ethanol in alkaline solution can break the resistance of the cell wall of the plant [80].

Results from other studies [81] show that the hemicelluloses obtained by ultrasound-assisted extraction seem to be more linear and less acidic with a relatively lower content of associated lignin, but with a higher molecular weight and a slightly higher thermal stability in comparison to hemicelluloses isolated by alkali without ultrasonic irradiation.

In recent research [82], Ma et al. studied the influence of simultaneous and sequence enzymatic pre-treatment with glucanase-G and xylanase-X on the efficiency of HCs extraction from paper-grade pulp using ionic liquid/water (IL/w) systems. Under optimized conditions, by enzymatic pre-treatments similar removal efficiencies are achieved, especially with ionic liquid/ water systems with high water content (25% and 20%). By simultaneous enzymatic treatment more efficient and selective HCs removal than sequential G–X and X–G treatments are obtained. Sequential extraction with ionic liquid/water-20 and ionic liquid/water-15 was demonstrated to be very efficient for HCs extraction without cellulose loss, allowing the xylose content of pulp to be lowered to 1.5%.

Supercritical water treatment of lignocellulosic is thought to be a promising method due to their advantages: short reaction time (several seconds) and no catalysts. It is reported that, by the supercritical water treatment of woody biomass, the fermentable sugars (glucose and mannose) can be recovered as cellulose and HCs-derived products in the water-soluble portion. Furthermore, the lignin-derived products are simultaneously recovered as the methanol soluble portion [83].

The selective separation of HCs from biomass include acids, water (liquid or steam), organic solvents and alkaline additives treatments, but the fractionalization and purification of the extracted hemicelluloses is sometime very difficult and remains the main challenge in this research area.

- Chemical Modifications of HCs

Hemicelluloses are hydrophilic polymers with extensive hydrogen bonds, so that the hydrophilicity limits the area of their industrial applications. The abundance of free hydroxyl groups distributed along the backbone and side chains make hemicellulose an ideal candidate for chemical functionalization. Chemically modified hemicellulose has high potential to be used to obtain materials with unique properties, which could improve the added value of the biopolymers. Therefore, chemical modification has been applied to enhance the function of hemicelluloses. This includes oxidation, reduction, esterification (e.g., acetylation, quaternization), etherification (e.g., carboxymethylation, alkylation).

By these treatments hydrophobic groups are introduced into the hemicellulose chain of HCs to improve the thermal stability, solubility in organic solvents, water resistance and rheological properties when are used in barrier coatings for packaging paper application [74].

The hydrophilic character of xylan-HCs can be modified as a function of the packaging end-use by ionic interactions with cationic/amphoteric/amphyphylic biopolymers (e.g., chitosan and cellulose derivatives) that can form polyelectrolyte complexes. For the paper packagings which require an equilibrium barrier (adsorption/desorption of humidity), the HCs coatings can be modelled by the use of inorganic filler (i.e., nanoclays). Additionally, the film-forming properties of hemicelluloses, highly desired for coating application, can be improved by adding of plasticizers (e.g., glycerol, sorbitol and xylitol) or by physical blend with other biopolymers such as alginate and carboxymethylcellulose [74,84–88]. The water resistance and barrier properties of polymers can be improved by cross-linking. However, hemicellulose cross-linking is mostly used to make hydrogels with limited water solubility, despite their ability to swell in water [89].

- Applications of HCs in Food Packaging

Literature reviews show the market volume of HCs increasing, and this class of polysaccharides has high potential to be very useful in many applications. Based on the studies on utilization of hemicelluloses from cereal straws, it is demonstrated that these polymers are a fermentation feedstock in platform of chemicals production (furfural, ethanol, acetone, butanol, xylose and xylitol). Furfural extracted from pentosanes of agriculture residues and hardwoods is an important chemical platform used as starting materials for nylon production. Xylitol and mannitol derived from xylan and mannan are high-value commercial products widely used as alternative sweeteners in food and pharmaceutical industry [90]. Others oligosaccharides compounds extracted from hemicelluloses have been used as prebiotics in food and feed industry [91–93].

In the paper industry, native or modified hemicelluloses have been used as additive for improving paper strength, retention aid, as binder in paper coatings or as wet reinforcement for cellulose nanocomposites [94,95].

Literature reviews show that the main applications of the HCs as barrier material are for stand-alone packaging films, with only a few of them being related to packaging paper coatings.

Generally, the HCs that preserve most of their side chains during extraction processes are highly water-soluble and have film forming ability [96]. It was reported films based on xylan/glucomannan provide good barriers against the oxygen and other gases, grease and aroma that enhance their potential for application on food paper packaging. However, the barrier and strength properties of HCs films are negatively influenced by wet environment [6,97].

When hemicelluloses are used as packaging materials, there are several disadvantages or issues influenced by their structural nature. These are: compatibility with the traditional plastics, thermal stability, mechanical properties and water vapor permeability [96].

Due to the numerous hydroxyl groups distributed along the HCs backbone and side chains, in their native form, hemicelluloses are highly hydrophilic. This results in low moisture barriers and a low protection level against water when are used in packaging materials. The poor mechanical strength and flexibility of hemicellulose-based materials are caused by their molecular weight and degree of polymerization (DP) in the range of 80–450. In addition to their low mechanical strength, hemicellulose films are usually very sensitive to moisture [98,99].

Based on its good affinity for cellulose, xylan hemicellulose is used as a strengthening additive for paper or biocomposites in combination with this. It can also be chemical modified to produce a new material with appropriate properties, such as hydrophobicity, thermal formability and ability for film forming. The last feature is necessary for xylan hemicellulose to improve self-supporting barrier films when used in food packaging. Thus, xylan is plasticized and hydroxypropylated, resulting in a water-soluble derivative with improved film-forming properties. The internal plasticization with hydroxypropyl groups can be combined with external plasticization using glycerol or sorbitol [84].

A commercial product based xylan HCs has been developed in Sweden (Skalax), which can be applied as a thin film to surface of paper-based materials by dispersion coating to provide barrier properties against oxygen, aroma, grease and mineral oil [24,100].

Other studies show that when are used as food packaging, the water vapor permeability of xylan based films are influenced by the low stretch ability and other components.

The most common applications of this material are as oxygen barrier coatings on food or packaging, where the coated products provide mechanical support (like paper). Additionally, xylan's hemicellulose can be used for edible packaging for low moisture food, coating on fruits or cheese.

However, even though the raw material for xylan hemicellulose is abundant, the presence of commercial product on the market and its large scale extraction are still limited [6].

In recent studies [101], functional HCs composite films with flexibility, thermoplasticity and UV-shielding ability were obtained using esterified HCs with vinyl benzoate reinforced with poly(vinyl alcohol) (PVA) and zinc oxide nanoparticles (ZnO). The films showed good flexibility and moderate water and oxygen barrier abilities. It was proven that by chemical modification, HCs improve their solubility and processability, and the addition of PVA and ZnO enhances the functional properties of the films, making them suitable and sustainable materials for application in the packaging field [101].

In addition, xylan HC was used as an additive in nanofibrillated cellulose-based nanocomposite films to improve the water absorption capacity. The results demonstrate the feasibility of hemicellulose, which acts as a plasticizer in NFC films and their potential application for the preparation of bioinspired nanocomposite films for food packaging [102].

Glucomannan hemicellulose exists in high quantities in wood from coniferous trees (spruce) and exhibits interesting film-forming properties. It can be extracted from process streams of newsprint paper mills or from fiberboard manufacturing. Glucomannan films have demonstrated good gas barrier properties, and therefore this HC has potential for use as components in barrier coatings for cardboard or paper for food packaging. Cross-linked glucomannan is soluble in water and organic solvents, has good water absorption properties, and could be a potential alternative as a "green" absorption material (gels, etc.). The addition of plasticizers resulted in increased film flexibility and moisture sensitivity. As plasticizers, glycerol, sorbitol and xylitol are used, and the biopolymers such as alginate or carboxymethylcellulose. In addition, by combination with other biopolymers, the mechanical strength and film resistance towards humidity increased [103].

The hydrophobic softwood galacto-glucomannan films with low moisture sensitivity and improved barrier properties have been prepared by benzylation (surface styrene grafting and lamination) using plasma and vapor-phase treatments [104].

### 2.1.3. Starch

Starch is widely available and inexpensive agricultural raw material and the most frequently present bioadditive in the paper industry, being widely used for both wet-end and surface/coatings applications. In their native form starch can be used as sizing agent (in size press of paper machine) while after chemical physical, and/or enzymatic modifications it can be used as coating additive due to its excellent film forming ability. Starch can be extracted from agricultural plant sources such as, potato, corn, wheat, tapioca and rice [24,105].

Their chemical structure consists of linear amylose crystalline polymer (poly-$\alpha$-1,4-D-glucopyranoside), and amylopectine (poly-$\alpha$-1,4-D-glucopyranoside and $\alpha$-1,6-D-glucopyranoside) a branched and amorphous polymer at variable ratios depending on the source (Figure 8) [6].

**Figure 8.** Chemical structure of starch: (**a**) amylose; (**b**) amylopectine.

Due to the brittleness of the native material with semi-crystalline nature, starch films does not have adequate flexibility and mechanical properties when are used in packaging. These properties can be improved by plasticization with other materials, chemical or physical modification, enzymatic treatments or composite combinations. Glycerol, sorbitol or xylitol, are typically plasticizers used for reducing the brittleness of starch films [106].

The water vapor permeability and oxygen barrier properties of starch-glycerol films can be improved by compounding with polycaprolactone (above 20% concentration) [107].

To reduce the intermolecular hydrogen bonds of polymer chains and to provide good stability and flexibility of thermoplastic starch, glycerol and sorbitol are used as plasticizers. However, due to the issues regarding the solution viscosity, film formation, and resistance to retrogradation (liquid to gel formation), the thermoplastic starch is less used in paper coatings.

The packaging properties of the starch films can be improved by use of citric acid and gelatine [6]. In their studies, Kumar et al. [108] show that the modified starch films with gelatine at low concentration exhibit remarkable improvement of the mechanical properties. High gelatine content led enhancement of the water vapor migration rate, solubility, swelling index and moisture absorption. Furthermore, SEM images for these starch films suggested that gelatine and citric acid improve the structural properties of the film, such as compactness, porosity, and homogeneity [108].

However, for paper coatings, the most common practice is chemical oxidation of starch by reduction of chain length and molecular weight of oxidized starch. This will reduce the viscosity of the coating solution. The acetylation reaction is one of the most interesting ways to decrease starch hygroscopicity. Thus, a high efficiency in paper coatings had the starch derivatives such as: acetylated starch, cationic starch and hydroxypropylated. Coatings with acetylated starch have been applied on kraft paper, with results in significant reductions of the water absorptivity and WVP (water vapor permeability) and improvement of barrier properties against gases and aroma compounds, maintaining the foods quality during storage [109].

To impart hydrophobicity to starch, physical treatments have also been used. In the literature, studies [110,111] have presented a way of reducing the water sensitivity of corn starch films via a sulphur hexafluoride ($SF_6$) plasma treatment. The obtained results confirm that the relationship between fluoride and sulphur incorporation is influenced by the plasma power. Thus, with the increase of plasma treatment time the initial contact angle is increased, reaching a value close to 140°. Moreover, this maximum hydrophobization exhibits a stable behavior in time.

The potential of starch in paper coatings is based on its ability to be a good carrier for compounds with barriers or active antimicrobial properties, such as mineral nanoparticles (ZnO, MgO, metallic ions, nanoclays) or nanofibrillated cellulose—to obtain composite coatings with good barriers or active antimicrobial properties for food packaging.

Using carboxymethyl starch and ZnO nanoparticles composite coatings, the optical properties (e.g., whiteness and brightness) of coated paper are improved. Furthermore, these coated papers showed excellent antifungal and UV-protecting properties, compared to bulk ZnO-coated paper. This is

explained by the slow dissolution of ZnO in wet environments, resulting in $Zn^{2+}$ ions that are active in microbes' immobilization. This is also important for enhancing paper life [112,113].

In other research [67,114], the nanocomposites based on NFC and silver nanoparticles were embedded as fillers in starch-based coating formulations and applied on paper surfaces to improve the antimicrobial activity, water vapor transmission, oil resistance, and strength of the coated paper.

Starch with swollen water and crosslinking nanocrystals/nanoparticles obtained by a co-extrusion process has potential for use in surface sizing and paper coating. Starch nanoparticles have many advantages in comparison with traditional dispersions of cationic or anionic starch. These exhibit higher bonding strength and lower suspension viscosity at relatively high solid contents (up to 30 wt.%) [115].

In their literature review, Johansson et al. [5] described another possibility to improve the starch hydrophobicity when used in paper coatings. This consists in the addition at starch suspension of mineral pigments (hyper platy nano clays) with high aspect ratios (highly plate particles). The coating color is then applied in thin layers on the paper and board surface with good results in water, oil and air barrier properties, and in maintaining the freshness of packed products.

### 2.2. Polysaccharides from Marine Biomass

### 2.2.1. Chitosan and Chitosan Derivatives

Chitosan is a polysaccharide [(β-(1-4)-linked D-glucosamine and N-acetyl-D-glucosamine], obtained by deacetylation of natural chitin, which presents unique features: it is a cationic bio-polymer, with hydroxyl groups able to develop hydrogen bonds like cellulose fibers; it exhibits film forming ability and antimicrobial properties; and it is biodegradable, biocompatible, and non-toxic. Chitosan was investigated as an antimicrobial active polymer for edible/biodegradable coatings on fruit/vegetables, and in combination with synthetic/biopolymers (e.g., PLA) to obtain antimicrobial films [116].

For applications in papermaking, chitosan was studied, especially during last two decades: a recent review [117] showed that from 115 references, only five are published before the year 2000. The studies cover various processes of papermaking systems: (i) wet-end applications, including control of coagulation/flocculation phenomena, process water treatment, internal additives for the improvement of mechanical strength or sizing effectiveness, or as dye fixative in the production of colored paper, etc. [118], and (ii) surface applications: in antibacterial coatings, as barriers to gases and grease, and intelligent paper-based packaging materials [119].

The high interest of researchers for chitosan as a bio-additive in papermaking is justified by its features: it is obtained from renewable resources; is biodegradable, non-toxic and biocompatible; has a structure similar to cellulose; presents natural cationic charge; and is effective for bacteria and fungi inhibition. However, up to now, chitosan application at a commercial scale in papermaking has not been considered [117]. The main cause of the limited applications of chitosan, especially in paper surface coatings, is its lack of water-solubility under neutral/alkaline environments, which is the common medium of water-based coating formulas [120].

In paper coatings, chitosan and their derivatives were considered for active-antimicrobial features proved in many research studies [121,122], as well as for their barrier properties against water vapor, oxygen, oils and grease [123]. In these applications chitosan can be used as aqueous solution or emulsion, using different coating methods or size press to be applied on paper surface [124].

The active-antimicrobial and barrier properties of chitosan can be improved by its combination with other polymers (i.e., PLA), nanofibrillated cellulose or inorganic nanoparticles (nanoclays, metallic oxides, silver nanoparticles) [125,126].

In their studies Coma et al. and Wang et al. [119,127] show that by combination of chitosan with propolis, the antimicrobial and antioxidant capacity of paper and cellulosic packaging materials were improved. Additionally, other research studies show that by incorporating a heat-sensitive pigment (anthocyanin-ATH) into chitosan dispersion and then applying it as a surface coating on card paper an

intelligent packaging indicator is obtained, which indicates temperature variations in a specific range by irreversible visual color changes [128].

In other studies [129], a combination of gelatine-chitosan was applied individually, in mixtures or via layer by layer techniques as edible coatings on fresh cut melons, as food models. It was found that edible coatings based on two components formulations demonstrate effective microbial (bacteria and fungi) spoilage inhibition.

In fact, the major portion of the research reported in the literature concerns chitosan dissolution in acid solutions of organic acids (acetic, lactic, citric) with pH~4.5, which creates troubles when applied in paper coatings, such as: non-uniform layer on paper due to deep penetration of the acid solution into the paper web and consequently, low barrier properties; acid-hydrolysis of cellulosic fibers and negative impact on paper strength, as well as unfriendly environments for the workers and coating equipment.

The presence of primary amines at the C-2 position of the D-glucosamine and hydroxyl groups gives to chitosan unique structural features, that allow specific chemical reactions to provide water soluble chitosan derivatives under neutral/alkaline pH, and to attach new functional groups for specific applications. In this respect, the acylation, alkylation, carboxyalkylation, quaternization, thiolation, sulfation, phosphorylation, attachment of carbohydrates and dendrimers, and graft copolymerization are chemical reactions applied with good and proven results for chitosan modification [130].

Extensive research has been performed in obtaining water soluble derivatives for biomedical applications [131]. However, the research on using water soluble chitosan in paper coatings is still in its infancy. The utilisation of chitosan-acetate (solution of chitosan in acetic acid) in papermaking was studied very early, while water-soluble chitosan derivatives (ChDs) are used much later being related on use of quaternized chitosan for improving of strength and printing properties of paper [132], and the use of carboxymethyl-chitosan (CCh) in old paper conservation [133].

The chemical modification of chitosan is via reactions of its primary amino function at C-2 and two hydroxyl functionalities, which are the main routes to obtain soluble derivatives (Figure 9). The chemical reactions such as acetylation, alkylation, quaternization, grafting and chelation of metals are developed at amino groups, while the O-acetylation (e.g., O-carboxymethyl, cross-linked O-carboxymethyl chitosan), H-bonding with polar atoms or grafting are given by the hydroxyl functional groups [134–136].

**Figure 9.** Structure of chitosan and possible reaction positions [115].

Remarkable results were obtained in obtaining ChDs with functionalities designed for the application in paper heritage conservation [3,137].

A few studies were also initiated on obtaining antimicrobial paper for food packaging [138,139]. The water soluble chitosan derivatives (alkyl chitosan—ACh, quaternary chitosan—QCh and carboxymethyl chitosan—CCh) and microfibrillated cellulose were used as barrier and mechanical strength additives in coating formulas for packaging paper grades. The results have shown that the water and water vapor barriers are improved by applied of ACh alone or in combination with MFC, the tensile strength properties (15–20%) and water vapor transmission rate (WVTR) (~30%) are enhanced by using CCh, and QCh has moderate effects on the water barrier and strength properties [138,139].

Alkyl chitosan (ACh) is obtained by reductive amination of chitosan free amino groups using aliphatic aldehydes with different alkyl chain lengths (octanal, decanal, dodecyl aldehydes) and substitution degrees [136]. Alkyl chitosan is a water soluble derivative, with a slight cationic charge under neutral pH, having an amphiphilic character, obtained by the introduction of hydrophobic alkyl groups. Hydrophilic sites provide affinity for the cellulose fiber surface, while the hydrophobicity could contribute to an improved barrier to water and water vapors of the paper, which is very important for food packaging paper.

Quaternized chitosan (QCh) is obtained by the O-and N-acylation reaction of chitosan with (3-chlor-2-hydroxipropyl)-trimethyl-ammonium chloride (Quat-188) [140]. Quaternized chitosan exhibits water solubility and cationic charge over the total pH range, high coagulation and complexation capacity, antimicrobial activity, hydrogen-bonding potential and high affinity for the cellulose fiber surface.

Carboxymethyl chitosan (CMCh) is synthetized by alkalization of the chitosan, followed by etherification with monochloroacetic acid (MClAc). The method leads to the occurrence of the both O- and N-carboxymethylation. The CMCh proprieties are affected by the reaction stoichiometry (MClAc acid:chitosan molar ratio) and other reaction conditions (time, temperature) as well as by chitosan properties [141]. Carboxymethyl chitosan has water solubility under neutral, alkaline and slightly acidic conditions, with quite stable properties of the solutions which result in high film forming ability and amphoteric character of the molecules, which confers high complexation capacity.

### 2.2.2. Alginates

Alginate is a polysaccharide usually available as salts of sodium and calcium of alginic acid and naturally present in brown algae. Advantageous properties such as film-forming ability, nontoxicity, biodegradability and biocompatibility are features that make alginate as one of the most promising and intensively studied biopolymers.

Chemically, alginates consists of (1-4)-linked β-L-guluronic acid (G) and α-D-mannuronic acid (M) (Figure 10).

**Figure 10.** Chemical structure of seaweed alginates.

Due to of the large amount of alginates and its derivatives already used as additives in the food industry, these biopolymers are considered to be safe for their use as functional barriers for food-contact materials. Moreover, different water-soluble alginate formulations are available on the market, which can be applied with conventional coating methods used in the paper and packaging industries [142–145]. However, there are only a few reports in the literature concerning the potential of alginate coatings to improve water barrier properties of coated paper, either used as sole polymer or in combination with other biopolymers (i.e., chitosan). Results show that this biopolymer is not able to reduce the water resistance of paper but has some synergistic effect when used in combination with chitosan [146,147].

In their research, Kopacic et al. [142] investigated the barrier properties and potential to reduce the migration and permeation oil saturated hydrocarbons and mineral oil aromatic hydrocarbons for papers coated with chitosan and alginate biopolymers. Both were applied using a draw-down coater onto two different paper grades from primary and secondary cellulosic fibers, respectively. The results

showed that for the coating on the secondary fibers substrate, a decreasing of at least 35% of the water-vapor transmission rate is obtained, and the overall migration of organic volatile compounds was successfully reduced by 70% and 84%.

The improvement of the water resistance of paperboard was obtained using the water-soluble sodium alginate combined with calcium chloride, by applying sodium alginate as the first layer on paper surface, followed by a post-treatment of dip-coating calcium chloride. Furthermore, the coated paper with sodium alginate exhibits excellent oil resistance via the application of only 5 g/m$^2$ of coating weight [148].

In recent research studies [149] composite coatings based on sodium alginate (SA)/sodium carboxymethyl cellulose and SA/propylene glycol alginate were used for the obtainment of fluoro-free greaseproof papers. The results showed very good kit values that met the requirement in food package applications (about value 9). The improvement of water resistance by decrease of surface energy and impact of polarity on oil resistance was obtained by a combination of propylene glycol alginate with lower polarity with sodium alginate. The mechanical properties of the coated paper were modified by penetration of coatings into the inter-fiber networks to some extent. A decrease of air permeability and the better oil resistance of the biopolymer coatings on fiber substrate were also obtained. For coatings with high surface energy, as the surface energy is dominated by the polar part, as the polarity strengthens, the repulsion to oil is higher. The reduction of surface energy can effectively improve the resistance for oil and water of paper.

## 3. Conclusions

Bio-polymer composite coatings for paper represent a sustainable alternative, providing the opportunity to obtain barrier properties (low oxygen and water vapor permeability) and specific functionalities for a fully protective food packaging.

The literature survey shows that polysaccharides are promising candidates to substitute petroleum based polymers in coatings for food packaging paper. These biopolymers are biodegradable, have high availability in nature, are non-toxic, and have film forming ability and good affinity for paper substrate with positive effects on the mechanical strength. Moreover, they can provide very good barrier to water, gases, aroma, and lipids, and serve as a biopolymer matrix for incorporation of active agents for paper functionalization. The main disadvantage of polysaccharides is their sensitivity to moisture that limits their large-scale utilization in barrier coatings for paper. The presented review shows that there has been intensive research regarding the chemical modification of polysaccharides, to introduce hydrophobic groups in their structure that improve water resistance and rheological properties when used in barrier coatings for packaging paper applications.

Therefore, the finding of an appropriate route for chemical modification—and the right combination of polysaccharides with other biopolymers, nanofibers or nanofillers—will generate interest for the development and application of these biopolymers in composite coatings for food packaging papers.

**Author Contributions:** Conceptualization, P.N.; Data curation, M.R. and P.N.; Resources, M.R. and P.N.; Writing—Original Draft Preparation, M.R.; Writing—Review & Editing, P.N.; Investigations, P.N. and M.R.; Supervision, P.N.; Visualisation M.R. All authors have read and agreed to the published version of the manuscript.

**Funding:** This research received no external funding.

**Acknowledgments:** The authors thank for support of the Research Centre for Environmental and Agriculture "Lunca", within Engineering and Agronomy Faculty in Brăila, "Dunărea de Jos" University of Galați, Romania.

**Conflicts of Interest:** The authors declare no conflict of interest.

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
