# Peer review of "Review on Polysaccharides Used in Coatings for Food Packaging Papers"

_coatings, doi:10.3390/coatings10060566_

Round 1

Reviewer 1 Report

It can be seen that authors have reviewed a lot of topic-related literature. Manuscript contains large amount of information about polysaccharides used as paper coatings, however the structure of paper must be improved.

Manuscript appears as non-connected facts from several publications, a lot of repeating facts and information. Review paper must have strict structure, combining several similar facts together, finding the things in common from different authors and also controversial facts have to be accented. Structure of manuscript is confusing, mainly incomprehensible.

Comments:

Line 6. Unnecessary Repeating words

Introduction. Bio-based polymers and biopolymers – more explanations, how and why you use these two terms. What are differences? What makes polymer bio or bio-based?

The language must be improved. Style and structure of text must be reviewed, there are misprints in text.

Paragraphs Line 91-94 and Line 95-99 seem repeating very similar information, it would be necessary to merge and rewrite them.

Line 117-118. Cellulose non-reactivity and structural properties depend more on crystalline regions, not amorphous as it is mentioned in manuscript. Amorphous regions are more achievable for chemical or any other exposure. Either explain your opinion more in details or review this part of the manuscript.

There is information about synthesis of cellulose ethers, but limited information about cellulose esters. Could you please explain your position?

Line 183. Cellulose nanocrystals don’t have fibrillar structure, so they can not be positioned as “type of fibrillar structure” of cellulose.

Producing of CNF can be realized not only using high-pressure homogenizers. Please expand description of CNF.

Confusing structure of hemicelluloses part. Applications first, mixed with general characterization. Then followed by extraction of HC from biomass.

Part 2.2. has better structure.

I suggest review your work properly, use native english speaking person review the language and style.

Reviewer 2 Report

General remarks

The work is written in an interesting way and such a synthetic analysis of the issue required a lot of work from the authors. And for that they deserve praise. I allowed myself to suggest only few small things that could be improved.

Polysaccharides as a way to address environmental concerns also recently appears when it comes to clothing production. This is particularly spectacular when it comes to bacterial cellulose which grows into dense almost ready for use sheet. Maybe for context it would be worth mentioning (cite) in this publication this issue described in the work: Cellulose 27, 5353–5365 (2020). https://doi.org/10.1007/s10570-020-03128-3. Especially because the authors do not mention anywhere in the whole work about bacterial cellulose. This type of cellulose is dramatically different from wood cellulose in terms of purification and processing.

Minor critical remarks

Line 93 Sentence:

„Are formed by the condensation of monosaccharide residues through hemi-acetal or hemi-ketal linkages, originate from from plant or marine organisms’ biomass”.

The authors should mention that polysaccharides are also synthesized in fungi (for example chitosan), yeast (some beta glucans) and (not on a scale that allows material use) mammalian cells (glycogen and glycosaminoglycans)

Reviewer 3 Report

The manuscript by Nechita et al. revises polysaccharides potential in paper industry for food packaging. The Review is well structured and deals with a hot topic, which will attract several readers. The introduction section describes the main theme of the paper, citing up-to-date literature. The following section 2 systematically presents the most abundant and promising carbohydrates to coat paper-based, food packaging systems. The conclusion section seems to be a repetition of the abstract and, in the present form, does not provide essential information to the reader. It should be improved adding the concrete role, in paper coating technology, of the main polysaccharides presented in section 2. Moreover, the critical comment on the advantages and current drawbacks of carboyhdrate-based coatings should be extended.   In order to improve the quality of the Review, I suggest the folloiwng minor revisions.   -Please check the manuscript to remove typos and grammar errors (e.g. lines 91, 93, 118, 135, 196, 225, 227, 332 ,464, 471, 499, 570, 588, 692). -Figures 1, 3, 6 and 9 are of low resolution. If possible, enhance their quality.

Round 2

Reviewer 1 Report

Dear authors,

Thank you for your effort!

It can be seen that you have carefully reviewed your work. All rewrited parts and corrections are appropriate. The clarity and structure of paper are increased. Thank you for answering and taking into account all comments made. I believe that after english editing it will be even better paper for publishing.

Author Response

Dear reviewer,

Thank you very much for your comments!